# Outcomes of Importance to Patients in Reproductive Genetic Carrier Screening: A Qualitative Study to Inform a Core Outcome Set

**DOI:** 10.3390/jpm12081310

**Published:** 2022-08-12

**Authors:** Ebony Richardson, Alison McEwen, Toby Newton-John, Ashley Crook, Chris Jacobs

**Affiliations:** Graduate School of Health, University of Technology Sydney, Chippendale, NSW 2008, Australia

**Keywords:** reproductive genetic carrier screening, core outcome set, patient perspective

## Abstract

There is significant heterogeneity in the outcomes assessed across studies of reproductive genetic carrier screening (RGCS). Only a small number of studies have measured patient-reported outcomes or included patients in the selection of outcomes that are meaningful to them. This study was a cross-sectional, qualitative study of 15 patient participants conducted to inform a core outcome set. A core outcome set is an approach to facilitate standardisation in outcome reporting, allowing direct comparison of outcomes across studies to enhance understanding of impacts and potential harms. The aim of this study was to incorporate the patient perspective in the development of a core outcome set by eliciting a detailed understanding of outcomes of importance to patients. Data were collected via online, semi-structured interviews using a novel method informed by co-design and the nominal group technique. Data were analysed using reflexive thematic analysis. Outcomes elicited from patient stakeholder interviews highlighted several under-explored areas for future research. This includes the role of grief and loss in increased risk couples, the role of empowerment in conceptualising the utility of RGCS, the impact of societal context and barriers that contribute to negative experiences, and the role of genetic counselling in ensuring that information needs are met and informed choice facilitated as RGCS becomes increasingly routine. Future research should focus on incorporating outcomes that accurately reflect patient needs and experience.

## 1. Introduction

Reproductive genetic carrier screening (RGCS) is a genetic test that allows prospective parents to determine if they are at increased risk of having a child with a recessive genetic condition, facilitating informed decision-making regarding how to proceed with their family planning. RGCS can range in complexity from haemoglobinopathy screening in routine prenatal care, through to expanded carrier screening of hundreds to thousands of genetic conditions in preconception or prenatal settings. Current consensus-derived practice guidelines recommend that RGCS is offered to all women planning a pregnancy or in their first trimester [1,2,3]. In a previous sequential systematic review of quantitative and qualitative studies on RGCS, we highlighted the heterogeneity of research to date and identified a need for standardised outcome reporting to inform evidence-based practice recommendations that can draw on a robust underlying literature [4,5].

The Core Outcome Development for Carrier Screening (CODECS) study aims to develop a core outcome set (COS) for studies on RGCS [6]. A COS is a minimum set of outcomes that should be measured and reported in all studies on a particular topic. There are three key stages of COS development: (i) reviewing current evidence, (ii) consulting with key stakeholders, and (iii) a consensus process to decide which outcomes are prioritised for inclusion. The CODECS study aims to ensure the perspectives of patients are strongly represented in the development of a COS. Previous COS studies on other topics have demonstrated that the inclusion of patients in a consultative process results in outcomes that would not have been suggested by health professionals alone [7,8]. They demonstrate that incorporating qualitative methods ensures that outcomes are meaningful for patients, provides a deeper understanding of why outcomes are valued and how they are prioritised by patients, guides the scope and language used to describe outcomes, and allows comparison of patient-derived outcomes with those from other sources such as systematic reviews [8]. 

Qualitative research in the development of a COS is evolving, and there is currently limited guidance on the best methods to utilise. A key challenge is how to approach eliciting outcomes with lay stakeholders, who may be unfamiliar with this concept [8]. Drawing on examples from the literature and utilising the theoretical frameworks of co-design and nominal group technique, we developed a novel approach to eliciting outcomes of importance to patients [9,10,11]. Co-design allows users to become part of the design team as “experts of their experience” [12]. While many COS studies have shown the value of co-design with patients, few have expanded on how best to utilise co-design principles to engage with patients when eliciting outcomes to include in the consensus process. Nominal group technique is a structured process used to achieve consensus amongst small groups [9,13]. We adapted key aspects of the nominal group technique for application in one-on-one interviews, including initial generation of ideas by participants without input or prompting from the interviewer, recording and discussion of each idea through a shared medium (virtual whiteboard), and prioritisation of ideas by participants. This structured approach allowed participants to produce descriptions of their experience in a way that enabled conversion into measurable outcomes. 

This study reports on the results of qualitative interviews with patient stakeholders designed to elicit outcomes of importance to prospective parents accessing RGCS. The outcomes identified herein will be added to the “long list” of outcomes collated from previous systematic reviews, and taken forward to the consensus process to determine which outcomes should be defined in a COS for RGCS. This study has two aims:*Aim 1:* To explore the themes underlying participant interviews and how these inform our understanding of outcomes that are important to prospective parents accessing RGCS.*Aim 2:* To explore the role of including qualitative consultation with patient stakeholders in the development of a COS.

## 2. Materials and Methods

This study was reported according to the consolidated criteria for reporting qualitative research (COREQ) [14]. Ethics approval was granted by the University of Technology Sydney Ethics Committee (UTS HREC ETH20-5179). 

### 2.1. Theoretical Paradigm

We approached this study through a constructivist paradigm, where the interaction between the researcher and participant, and the influence that has on the resulting data, is viewed as an essential component that drives the co-creation of knowledge [15]. To engage in self-examination and consider how the researcher’s knowledge, assumptions or biases influenced the data collection, ER wrote reflective notes after each interview and throughout the analysis process.

### 2.2. Recruitment and Patient and Public Involvement

Individuals or couples who accessed RGCS to inform their reproductive decisions were eligible to participate in this study. For the purpose of this study, individuals and couples undertaking RGCS will be referred to as patients; however, we acknowledge that these will be largely healthy adults, most of which will not go on to require significant medical follow-up as a result of their carrier screening results. Participant groups were defined by two characteristics: their level of risk prior to RGCS (a priori) and their level of risk following results (a posteriori). Average a priori risk was defined as the participant having no existing health concerns or family history to indicate an increased risk of being a carrier. Increased a priori risk was defined as the participant having an existing factor such as ethnicity with a known increased carrier frequency or a known family history of a recessive genetic condition. A posteriori risk was grouped into either low or increased reproductive risk. Low reproductive risk results were defined as those where neither member of a reproductive couple were found to be carriers of the same genetic condition, or where one member of a reproductive couple was found to be a carrier of an autosomal recessive genetic condition but their reproductive partner was not a carrier of the same condition. Increased reproductive risk results were defined as those where both members of a reproductive couple were found to be carriers of the same autosomal recessive genetic conditions, or where the female reproductive partner was found to be a carrier of an X-linked recessive genetic condition. 

Different outcomes were expected to arise between participants based on their a priori and a posteriori risks, as well as the setting and context within which they accessed RGCS. As such, we utilised a broad passive social media approach to recruit a diverse international sample of participants with a range of RGCS experiences to capture a variety of outcomes to consider for inclusion in a COS set. An expression of interest to participate in the research was circulated through online parenting forums, Twitter and a Facebook group for carriers of genetic conditions. Respondents were directed to a brief survey to confirm eligibility, provide contact information and indicate if they would prefer to participate in a one-on-one interview, focus group or did not have a preference. Once eligibility was confirmed participants were contacted via email and a meeting time arranged. 

### 2.3. Inclusion and Exclusion Criteria

Individuals and couples were eligible to participate if they had undertaken RGCS and received a result. Those who had not yet received results were excluded. Health professionals were excluded to ensure that data reflected lay experiences. 

### 2.4. Participant Selection

Purposive sampling was used, aiming for equal representation across a priori and a posteriori groups, and diverse international representation. We approached sample size through the lens of recent commentaries that highlight the problematic nature of aiming for “saturation” as an end-point for recruitment [16,17]. We instead adopted the concept of theoretical sufficiency, which seeks the point at which the researcher has sufficient depth of understanding to address the study aims [16]. In the context of this COS development study, theoretical sufficiency was represented as the point at which a range of patient experiences that encapsulated outcomes of importance were captured. The goal was not to identify all possible outcomes, but rather those of most importance that warrant consideration for inclusion in a COS. Data collection and analysis were performed concurrently. Data collection ceased when sufficient richness of information was achieved. 

### 2.5. Data Collection

In-depth semi structured interviews were conducted remotely via Zoom [18] by ER between June and October 2021. Interviews were audio- and video-recorded. Basic demographic information was collected at the beginning of interviews. We used recent guidance on reporting race and ethnicity in medical and science journals to define categories based on participant’s self-reported ethnicity [19]. We developed an interview schedule that engaged participants in a discussion about their experience and allowed them to conceptualise outcomes that were appropriate to capture it. The interview guide was informed by our sequential systematic review [4,5] and developed iteratively with two patient representatives (available in Appendix A). The interview guide was broken into four sections, becoming increasingly specific as the interview progressed (Figure 1): Narrative exploration—participants were asked open questions about their uptake of RGCS and prompted to tell the story of their experience.Word association exercise—participants oriented themselves within the narrative they had just described and wrote down words that came to mind to describe their experience. The RGCS process was divided into four time frames; pre-test counselling and deciding to access RGCS, waiting for results, receiving results and the immediate follow-up period, and the long-term perspective. Participants used pen and paper to record their words for each time frame and were given up to 5 minutes to write down as many words as they could think of. A blank virtual whiteboard was shared with each participant and their chosen words were recorded.Exploration of generated words and eliciting of outcomes—participants expanded on each word they had written, and why they thought that word had come to mind. Through this exploration, the interviewer guided the participant in conceptualising how these words could be converted into measurable research outcomes.Prioritisation exercise—participants considered the outcomes they had discussed with the interviewer and their importance, ranking the top 3 that they considered crucial for research to explore.

### 2.6. Data Analysis

Interviews were transcribed verbatim and analysed using reflexive thematic analysis [20]. The transcripts were not returned to participants for corrections, as this was deemed unnecessary due to the collaborative process of the interviews. ER and CJ independently coded two transcripts and compared and discussed codes. ER then coded the remaining interviews. CJ reviewed the codes for the remaining interviews and discussed the approach and reasoning behind the codes. Data was stored and managed using NVivo software [21]. 

An inductive approach to coding was adopted, initially utilising semantic codes that closely reflected the participants’ own words. Codes were developed iteratively and those with close semantic similarity were merged and associated with an overarching outcome code. The virtual whiteboard generated with each participant during their interview was compared to the codes derived from their transcript to ensure that all elicited outcomes had been captured. 

We used a deductive approach to map the outcomes to an existing taxonomy previously developed for this study (available in Appendix A), and to an overarching taxonomy defined by the COMET initiative [22]. This analysis resulted in a list of outcomes that were elicited from patients, grouped into outcome domains, which was directly comparable to outcome domains reported in a previous sequential systematic review [4,5]. Outcomes were collated to determine a final long list that would be included in the consensus process for development of a COS and assessed for outcomes that were uniquely identified by patients in this study. 

Outcome domains were analysed to develop themes to answer the question “which outcomes of RGCS are most important to patients, and most accurately capture their needs and experience?”. Illustrative quotes are denoted by study ID, reproductive risk, setting of RGCS access and country of residence.

## 3. Results

### 3.1. Participant Characteristics

The majority of participants (*n =* 10) that responded to the EOI indicated a preference for one-on-one or couple interviews, while the remainder had no preference (*n =* 5), For consistency, and due to time-zone complexities, it was decided that all participants would be interviewed as opposed to convening a focus group. Interviews were conducted with 15 participants (nine individual interviews, and three couple interviews). The majority of participants were female (*n =* 12, 80%), and all males participated in a couples interview with their partner. Due to the social media approach utilised for recruitment, non-participation was unable to be assessed. The average interview duration was 61 min (range 38–84 min). The mean age of participants was 32 (range 25–46). All participants had undergone RGCS within the last 5 years. Table 1 describes the sample characteristics.

### 3.2. Aim 1: To Explore the Themes Underlying Participant Interviews and How These Inform Our Understanding of Outcomes That Are Important to Prospective Parents Accessing RGCS

Four core themes were identified from participant interviews: the importance of pre- and post-test genetic counselling for patient experience, psychological wellbeing in increased risk couples, challenges and barriers facing patients, and perceived utility of RGCS from the patient perspective. These themes highlight outcomes of RGCS that are most important to patients, and which may warrant focus in future research.

#### 3.2.1. Theme 1: The Importance of Pre- and Post-Test Genetic Counselling for Patient Experience

For the purpose of this study, genetic counselling is considered a communication process that can be performed by a range of health professionals including GPs, midwives, obstetrician gynaecologists and maternal fetal medicine specialists, as well as specially trained genetic health professionals such as genetic counsellors and clinical geneticists [23]. All participants described pre- and post-test genetic counselling with their health provider as a crucial aspect of their RGCS experience and identified a number of key goals health providers should strive for.

All participants discussed the role of genetic counselling in promoting a sense of reproductive empowerment, by facilitating the provision of information regarding reproductive risk and assisting with the comprehension and understanding of the implications of results.


*“Receiving results was two-fold; it was feeling informed and empowered by being informed.”*
—ID-9, increased risk couple, RGCS following fetal loss, Canada.

Participants discussed their perception of whether information needs had been met at various stages throughout the RGCS process. Many thought that there was room for improvement in pre-test genetic counselling to ensure that patients adequately understand what RGCS is and the possible implications of results.
*“I would say I was half informed, like I think that our doctor and the representative from the testing place…could have done a little bit of a better job explaining exactly what was being tested for.”*—ID-1, increased risk couple, proactive RGCS in the prenatal setting, USA.
Others found genetic counselling to be informative and that their needs were met.


*“We had really good communication from the genetic counsellor who handled everything…they were really thorough with how they explained everything.”*
—ID-10, low risk couple, proactive RGCS in the preconception setting, Australia.

Participants commented on the importance of feeling supported and understood during genetic counselling. 


*“We had the call for the results and we were able to ask questions there, but then she also gave us her email and said if you do have any questions just send us an email and yeah, so it was very supportive.”*
—ID-4, low risk couple, proactive testing in the preconception setting, Australia.

Nuances of different settings were apparent, with participants who accessed RGCS as part of an IVF work up or through midwifery screening programs in routine prenatal care commenting that more information was needed. 


*“I think in general they should maybe sit down with you at the start, even with a pamphlet and just let you know [the details]... I think that at the start, it would be better if they just were a bit clearer.”*
—ID-3, low risk couple, RGCS in the IVF setting, NZ.

Making an informed choice was also discussed by participants who accessed RGCS in IVF or routine prenatal care settings. In these settings screening was experienced as less of a choice and participants put trust in their healthcare providers to decide what testing was appropriate.


*“It almost wasn’t really given as a choice for genetic testing for us, the doctor kind of just was like ‘you guys should do this’, and we were like’ okay if you say so’. It wasn’t a ‘we want to do this’, it was ‘you should do this… Yeah, and you trust your doctor to know what they’re recommending to you, and just say OK great.”*
—ID-1, increased risk couple, proactive RGCS in the prenatal setting, USA.

While many types of information were discussed, practical information about next steps when receiving an increased risk result were a focus for many participants, with some noting that they were not equipped with all the information they needed.


*“Most of all is I want to know what my plan is now and I don’t think I had information on providers in the future and like what they do, what the process needs to look like if I want a natural route or an IVF route, like, I had to figure out all of that on my own, like I said, I still don’t fully comprehend exactly all the providers that I need to touch on if I went for a natural pregnancy.”*
—ID-8, increased risk couple, RGCS following fetal loss, USA.

#### 3.2.2. Theme 2: Psychological Wellbeing in Increased Risk Couples

Nine participants, including two couples, faced an increased reproductive risk when planning future pregnancies and provided insight into the long-term psychological impact of an increased risk result. Participants described grieving the loss of an expected pregnancy journey, and those who had also experienced a fetal loss were able to recognise an evolution from grieving the loss of a child to a prolonged grieving of their expected future. 


*“So you weren’t even losing the pregnancy and your child, but you’re losing all the stuff you’d mentally planned for…it’s a lot more than just grieving a child or the baby, it’s the whole life that you’d kind of dreamed up.”*
—ID-12, increased risk, RGCS following fetal loss, NZ.

Medicalisation of the journey to a healthy pregnancy was described by many participants, whether they accessed IVF with PGD or conceived naturally and undertook prenatal diagnosis; in all instances participants described the loss of spontaneity and joy around early pregnancy. 


*“It’s gone from trying to conceive naturally, which was very fun, to IVF which is very not fun.”*
—ID-2, increased risk couple, proactive RGCS in the preconception setting, Australia

The loss of time leading to the goal of a healthy pregnancy was also a source of grief for several participants.


*“There’s the loss of what we thought was, you know, ‘the’ pregnancy and then loss of time… in my mind, I’m thinking and processing the fact that like we’re ready to start our family, but there’s still so many steps to take before that.”*
—ID-1, increased risk couple, proactive RGCS in the prenatal setting, USA.

#### 3.2.3. Theme 3: Challenges and Barriers Facing Patients

When relaying their experience of accessing RGCS, all participants discussed barriers that negatively affected their experience. Many participants found it challenging to navigate the practical aspects of access and the complexities of the social context in which RGCS takes place. 

Cost and convenience of the process were frequently discussed, and how these may lead to a more motivated and higher socioeconomic group accessing RGCS.


*“Because yeah, there’s quite a big cost to all this testing as well in New Zealand and our obstetric care is public funded for the midwife system and so people don’t expect to pay a cent... when these opportunities aren’t offered or only offered at a cost, suddenly this rich versus poor barrier is put in place.”*
—ID-12, increased risk couple, RGCS following fetal loss, NZ.

A small number of participants felt that their provider had struck a balance with cost and convenience that facilitated their uptake of screening. 


*“Yeah, I think accessibility is the biggest thing. Like for example, most people would [look up] genetic tests and just click the first result and see $1500 and a big pdf where you have to go to the doctor and do that, do this. It’s just too much effort. Whereas [the test we accessed] is a lot more compelling…and affordable… you know in this on demand generation, order online, spit in this tube, and send it back and everything is done online, that is really good.”*
—ID-11, low risk couple, proactive testing in the preconception setting, Australia.

Many wished that there was more awareness of RGCS to facilitate it being offered to them preconception and supported this being a priority in the future. 


*“The regret is that we waited so long to start the process of finding out the disorder. And then you know maybe we would have done that before [the affected pregnancy] had it been offered.”*
—ID-1, increased risk, proactive RGCS in the prenatal setting, USA.

Social barriers and stigmatisation were a focus for many increased risk couples, highlighting a need for research and recognition of the social context in which patients are experiencing RGCS in order to be prepared for the challenges they may face. 


*“I feel more guarded about telling people that I’m expecting again. Because on either end there’s judgement.”*
—ID-9, increased risk couple, RGCS following fetal loss, Canada.

#### 3.2.4. Theme 4: Perceived Utility of RGCS from the Patient Perspective

All participants identified reproductive empowerment as the key outcome that represented their perceived utility of RGCS. 


*“Knowledge is power…by having that knowledge you’ve got the power to, and the confidence, to make future decisions without having really any worry in terms of genetics. There are always obviously random variations but in terms of on paper, I feel empowered.”*
—ID-11, low risk couple, proactive testing in the preconception setting, Australia.

Reproductive empowerment was a concept that was expressed regardless of risk, with increased risk couples feeling empowered to explore options, and low risk couples feeling empowered to proceed with natural conception.


*“That was as good as it could have been pretty much...I’m just happy that we could start planning and all that fun stuff...move forward without making any major changes to our plans.”*
—ID-10, low risk couple, proactive RGCS in the preconception setting, Australia.

The ability to inform reproductive decisions was also perceived as a key outcome of RGCS by most participants and was conceptualised as being complementary to their feelings of empowerment.


*“[The results] allow us to make data-backed or logical decisions for future pregnancies…we know we have three out of four shots of having a healthy pregnancy and so for us, compared to the IVF route, I feel like we can make decisions as long as they align with our emotional wellbeing to move forward with trying to conceive a healthy child naturally.”*
—ID-6, increased risk couple, RGCS following fetal loss, USA.

### 3.3. Aim 2: To Explore the Role of Including Qualitative Consultation with Patient Stakeholders in the Development of a COS

#### 3.3.1. Word Exercise and Eliciting Outcomes

Participants generated an average of 13 words (range 8–16) during the word association section of the interview. Through exploration of each word, an average of 16 outcomes (range 11–32) were elicited from each interview. In total, 55 unique outcomes were identified across the 15 interviews. Thirty-seven of these outcomes overlapped with outcomes identified from our previous sequential systematic reviews of quantitative studies [4] (*n =* 8), qualitative studies [5] (*n =* 13), or both (*n =* 16); and 18 were new outcomes.

In the twelve interviews that were conducted, the most frequent outcomes were “genetic counselling promoted reproductive empowerment” (*n =* 12, 100%), “patient-reported confidence/empowerment related to reproductive decisions” (*n =* 12,100%), “factors influencing access and uptake of RGCS” (*n =* 9, 75%), “genetic counselling provided sufficient information to meet patient needs” (*n =* 9, 75%), “patient-reported anxiety” (*n =* 9, 75%), “pre-test perceived risk of a carrier finding” (*n =* 7, 58%) and “grieving the loss of an expected pregnancy journey and medicalisation of future pregnancies in increased risk couples” (*n =* 7, 58%). These outcomes are informed by and correspond to the previously described themes generated from this study.

#### 3.3.2. Outcome Domains

Outcome domains, hereafter referred to as CODECS domains, were previously defined during data analysis for our sequential systematic review [4,5]. Definition of the domain and grouping of outcomes were developed iteratively by ER and AC and taken to the study management group (CJ, AM, TNJ) for final review and consensus. Twenty-six CODECS domains were defined. 

Outcomes from this qualitative study mapped to 18 CODECS domains (Figure 2). All outcomes mapped to existing CODECS domains; therefore, no new outcome domains were identified. Nine overarching COMET taxonomy outcome domains were represented: delivery of care (*n =* 12, 100%), cognitive functioning (*n =* 12, 100%), emotional functioning/wellbeing (*n =* 10, 83%), social functioning (*n =* 7, 58%), personal circumstances (*n =* 7, 58%), need for further intervention (*n =* 6, 50%), pregnancy, puerperium and perinatal outcomes (*n =* 5, 42%), congenital, familial and genetic outcomes (*n =* 2 17%), and perception of personal health (*n =* 1, 8%). 

#### 3.3.3. Prioritisation Exercise

The most frequently prioritised outcomes ranked by patients were “patient-reported confidence/empowerment related to reproductive decisions” (*n =* 10, 67%), “genetic counselling provided sufficient information to meet patient needs” (*n =* 8, 53%), “offering RGCS preconception is preferred (*n =* 6, 40%), “patient-reported anxiety” (*n =* 5, 33%) and “barriers and facilitators influencing patient experience of RGCS” (*n =* 5, 33%).

## 4. Discussion

Patients are ideally placed to identify outcomes that capture their needs and experiences. In this qualitative study, we sought to understand what outcomes are considered important to patients who undertake RGCS. The themes and outcomes we describe have implications for future research and for the development of a COS that is inclusive of the patient perspective.

Participants highlighted informed choice and information needs as key areas for improvement in their pre- and post-test genetic counselling experience. When considering informed choice, it was clear that participants experienced routinisation of RGCS in certain settings. The concept of routinisation has been explored broadly in the prenatal literature, especially in non-invasive prenatal screening [24] and has also been highlighted as an area of concern regarding the societal impact of RGCS [25,26]. Whilst there are many aspects and definitions of routinisation, most pertinent to these interviews was how healthcare providers frame RGCS. Healthcare providers often have the patient’s trust and are assumed to know best; therefore, when they present RGCS as routine, there is potential to undermine the ability of the patient to make an informed choice about whether to accept screening [27]. This can lead to a lack of deliberation regarding the implications if found to be at increased risk [24]. Routinisation may also result in patients feeling pressured to accept RGCS and feeling disempowered in their decision-making, and may affect societal perceptions of the acceptability of declining RGCS or not intervening to prevent the birth of an affected child [28]. Considering that patient-centred care is a key goal of genetic counselling, addressing routinisation and how it may negatively impact the patient experience of RGCS needs further exploration. Whilst some literature has explored this topic, little experiential evidence is available to guide appropriate implementation of RGCS as it becomes increasingly available to the general population [25,29,30]. 

Information provision is an important component of informed choice, with a high degree of variability described by patients in this study. Participants discussed their perception of whether their information needs were met, and areas where more information was needed, and disclosed how they were provided information ranging from verbal discussions and pamphlets to online education. A recent publication documenting the development of a decision aid highlights many of the aspects of information provision that patients discussed in our interviews [31]. An ACMG practice resource provides recommendations on the information that should be provided during pre- and post-test genetic counselling and reflected both where information provision was good and where it was lacking for our participants [32]. These two examples of evidence-based resources to guide healthcare providers involved in genetic counselling for RGCS are valuable additions to the implementation of RGCS. The outcomes identified in this study, in concert with recent publications that document the key patient information components, set a strong foundation for future studies to accurately assess if patient information needs are being met and informed choice facilitated. 

In our previous systematic review of quantitative studies [4], we highlighted the heterogeneity in psychological outcomes measured in studies on RGCS. There were a wide variety of psychological outcomes chosen by researchers, anxiety and worry being principal among these. However, no evidence exists for why these patient-reported outcomes were chosen and if patients were consulted to inform the most meaningful outcomes to assess. In this qualitative study, as well as in a systematic review of previous qualitative studies on RGCS [5], we identified grief as a relevant psychological outcome in increased risk couples. The concept of grief as an emotional implication of genetic diseases has been explored across the genetics literature [33] and it has been identified as an important component of the adjustment period for increased risk couples following RGCS [34]. However to our knowledge, grief has not been incorporated into any quantitative evaluations of RGCS to date. In this study, grief was evident across increased risk couples and related to tangible losses such as the termination of affected pregnancies and fetal losses, as well as less tangible concepts such as the expected pregnancy journey, time and adaptation to a medicalised process to a healthy child. Grief, bereavement and traumatic stress as a result of reproductive losses have been explored in obstetric and fertility settings that are innately intertwined with RGCS. Reproductive loss has been defined as “any absence of innate function, missing of a promised child, or a tangible loss surrounding the natural human cycle of propagation”, and covers a wide range of losses [35]. Reproductive losses experienced by couples identified as increased risk through RGCS are broad and may include the experience of prenatal diagnosis leading to termination of an affected pregnancy, undertaking IVF/PGD with unsuccessful cycles or additional fertility challenges, loss of planned future children if results inform the decision to reduce planned family size or loss of being biologically related to future children if they elect to use donor gametes; with RGCS being the inciting event that leads to the experience of these reproductive losses. As such, there is a need to investigate grief in the setting of RGCS to ensure that increased risk couples are supported in their journey to a healthy child. 

The experience of complex grief is a potential adverse outcome of RGCS. While adverse outcomes are unlikely to be seen in the majority of patients undergoing RGCS, similar to other areas of genetics it is likely that a small percentage of patients will be affected and require additional supports. As the number of individuals from the general population accessing RGCS grows, even a small percentage may equate to a significant number of patients. Individuals with increased risk results are most likely to experience complex and ongoing psychological impacts following RGCS, and this group is significantly under-explored in the literature. Only two studies from our systematic review of quantitative studies [4] measured psychological wellbeing in a cohort that included increased risk couples (combined sample size *n =* 11). In comparison, five studies from our systematic review of qualitative studies [5] included increased risk couples (combined sample size *n =* 50). However, the concepts from such qualitative work are yet to be translated into measurable outcomes used in the evaluation of RGCS. Training in grief and loss is considered an important aspect of genetic counselling practice [36]. A recent study explored the role of genetic counsellors in mitigating complex or prolonged grief following termination of pregnancy for fetal anomalies and identified a major role for genetic counsellors in facilitating adaptive coping [37]. Future research aiming to gain a better understanding of grief related to reproductive losses in the setting of RGCS will ensure that health providers can provide support and promote effective coping in increased risk couples. A clear understanding of the frequency and severity with which increased risk couples may experience complex grief is needed if we are to develop evidence-based practice recommendations based on the assumption of negligible harms. 

We did not specifically seek to explore barriers in this study, but the prevalence with which participants raised barriers highlights that these are evident to patients undertaking RGCS and frequently impact their experience in negative ways. Participants identified three key areas: cost and accessibility, awareness amongst primary care providers and the community regarding RGCS in the preconception setting, and broad social factors that present both practical and psychological challenges for patients (e.g., entrenched societal views of termination and stigmatisation). A recent systematic review exploring barriers and enablers from the practitioner perspective captured many of the same barriers identified in this study, highlighting a need for further research informed by implementation science and behaviour change theory to address them [38]. Recent commentaries from Australia [25], the US [39] and the Netherlands [40] all contribute to this growing recognition of the social context, barriers and challenges that are key to consider in the implementation of RGCS. An important aspect of the barriers discussed by participants was timeliness. Cost and accessibility meant that many patients delayed or procrastinated about screening, and all participants that had accessed RGCS prenatally or following a fetal loss expressed regret that RGCS had not been accessible preconception. Those that had accessed termination of pregnancy perceived that a preconception offer could have spared them that experience and subsequent stigmatisation based on their reproductive decisions. A preference for patients to be offered RGCS preconception is prevalent across the literature on this topic [38,41], and will likely be a continued focus of research to address barriers and inform best practice as RGCS becomes widely implemented. 

The concept of empowerment is recognised as a significant outcome of RGCS and is frequently framed as a primary goal [42,43,44,45]. This study provides valuable insight into what patients consider the primary goal of RGCS and their perceived utility of RGCS. Participants defined reproductive decisions made by couples accessing RGCS as an important outcome and reflected that RGCS provided the information needed to make such decisions. They further elaborated on a second outcome of empowerment to reflect how equipped couples felt to make these decisions. The outcome of reproductive empowerment was the most highly prioritised outcome from these interviews. McAllister et al. have established empowerment as a key outcome of clinical genetic services [46,47], work which subsequently informed the development of an empowerment scale for use across a range of genetic settings [48]. Lewis et al. expanded on this work and, similar to our findings, identified reproductive empowerment as the main outcome of RGCS [49]. Despite this existing work, no quantitative studies published to date have evaluated empowerment using a patient-reported outcome measure in an RGCS cohort. Instead, informed decision-making underlies the concept of empowerment in current literature, with an assumption that RGCS empowers patients by providing information that they can use to make reproductive decisions in line with their estimated chance of having an affected child and their values regarding this. There are some issues with this approach though. Informed choice has been evaluated predominantly regarding uptake, with studies aiming to determine whether patients make informed decisions to access RGCS. The multi-dimensional measure of informed choice (MMIC) is most often used, assessing patient knowledge, attitude to testing and whether behaviour (i.e., the choice to accept or decline testing) is in line with their knowledge level and attitude [50]. This pre-test approach ensures appropriate access to RGCS but does not capture informed decision-making following results. Post-test evaluations aimed at capturing empowerment assess intended and actual reproductive decisions made by prospective parents based on RGCS results. These current pre- and post-test approaches ground their conceptualisation of empowerment in information and behaviour, but these are only two aspects of a more complex construct. McAllister et al. define empowerment as a construct that encapsulates five concepts: (1) Decisional control—being able to make important life decisions in informed ways. (2) Cognitive control—having sufficient information about the condition, including risks to oneself and one’s relatives, and any treatment, prevention and support available. (3) Behavioural control—being able to make effective use of the health and social care systems for the benefit of the whole family. (4) Emotional regulation—being able to manage one’s feelings about having a genetic condition in the family. (5) Hope—being able to look to the future with hope for a fulfilling family life, for oneself, one’s family and/or one’s future descendants [48]. The latter two components, emotional regulation and hope, are of particular relevance to this study. Participants described empowerment as a long-term or overarching outcome of RGCS, having ongoing relevance far beyond the immediate post-test period. Learning to cope with the implications of their results, manage their psychological wellbeing over time and have hope that they are working towards a healthy pregnancy underlined participant discussions around empowerment. While some aspects of empowerment have been captured in the literature to date through a focus on informed choice and reproductive decisions made by increased risk couples, there is nuance that is lost without considering the other aspects that make up the construct of empowerment. We have also previously highlighted timeliness as a component of patients’ perceived utility of RGCS [5]. Similarly, participants in this study had a preference for preconception offers as a way to further bolster empowerment. There is currently no consensus definition of the utility of RGCS; hence, there is an opportunity to incorporate the concepts we have identified in future studies. The clear conceptualisation of utility encompassing both the clinical aspect of reproductive decisions and the personal aspect of empowerment will ensure a truly patient-led and ethically-oriented definition of utility for the purpose of a core outcome set.

This study was informed by evidence from previous COS development studies that showed the valuable contribution of patients to the outcomes identified [8,51]. Consistent with these, we identified eighteen outcomes that were unique to patient interviews and would not have been identified from a systematic review alone. We also gained detailed and nuanced insight into how patients conceptualise outcomes of their RGCS experience, which highlighted clear gaps in current evidence and can inform future research. We report a novel methodology for eliciting outcomes from patients that facilitated this rich depth of understanding. This study supports current evidence on the importance of incorporating patient stakeholder consultations in a COS development study to ensure that outcomes are meaningful to patients, and perhaps more significantly, why these outcomes are important.

## 5. Conclusions

In consultation with patient stakeholders, we elicited a detailed understanding of outcomes considered important to patients undertaking RGCS. Grief and empowerment were highly valued outcomes that have not been assessed in studies of RGCS to date. Patients experienced barriers and challenges related to the societal context in which RGCS occurs, and experienced negative repercussions of routinisation that compromised informed choice. This study highlights that patients can be active partners in identifying meaningful outcomes of a health intervention and identify outcomes that may not have been considered without their input. To achieve patient-centred and ethical implementation of RGCS, there must be uptake of outcomes that matter to patients in future research.

## 6. Limitations

Participants responded to a broadly distributed EOI; therefore, there is a chance of selection bias. In addition, many of our participants had experienced fetal loss and we acknowledge that psychological outcomes may be overrepresented. The majority of participants were female, with three male partners participating in a couple interview. This may reflect upon recognised differences in the role of reproductive partners in decision-making regarding family planning and pregnancy. There is a chance that different outcomes may be considered important by male reproductive partners. As the purpose of these interviews was to elicit as many outcomes of potential significance as possible, any bias in participants may be beneficial in identifying outcomes that are not evident in other samples. Any overrepresentation will also be mitigated in subsequent consensus steps of the CODECS study, as the frequency with which outcomes have been seen does not influence whether they are presented to stakeholders for consideration for inclusion in the COS. In aiming for a diverse sample we achieved some degree of representation across different ethnic groups; however, all participants were from developed, English-speaking countries and were highly educated. Further work is needed to ensure that outcomes included in a COS are generalisable and representative of the needs across all populations accessing RGCS.

## 7. Research Team

ER (female) is a PhD candidate and associate genetic counsellor with training in qualitative research methodology, who at the time of data collection had 7 years’ experience discussing genetic health conditions and genetic testing with patients across clinical, research and laboratory settings. Participants were aware of the professional background of the interviewer. CJ (female) is a senior lecturer in the genetic counselling program at the University of Technology Sydney (UTS), a registered genetic counsellor and a registered nurse with qualitative research experience. AM (female) is an associate professor in genetic counselling at UTS, led the establishment of the Master of Genetic Counselling program at UTS and is a registered genetic counsellor with qualitative research experience. TN (male) is a professor of psychology at UTS, and an endorsed clinical psychologist with qualitative research experience. AC (female) is a PhD candidate and certified genetic counsellor with training in qualitative research methodology.

## Figures and Tables

**Figure 1 jpm-12-01310-f001:**
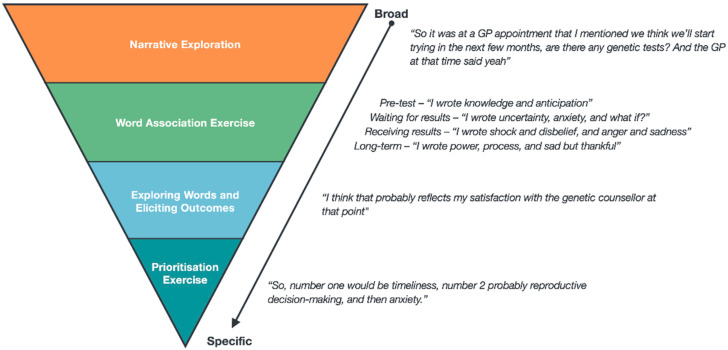
Interview schedule overview with examples.

**Figure 2 jpm-12-01310-f002:**
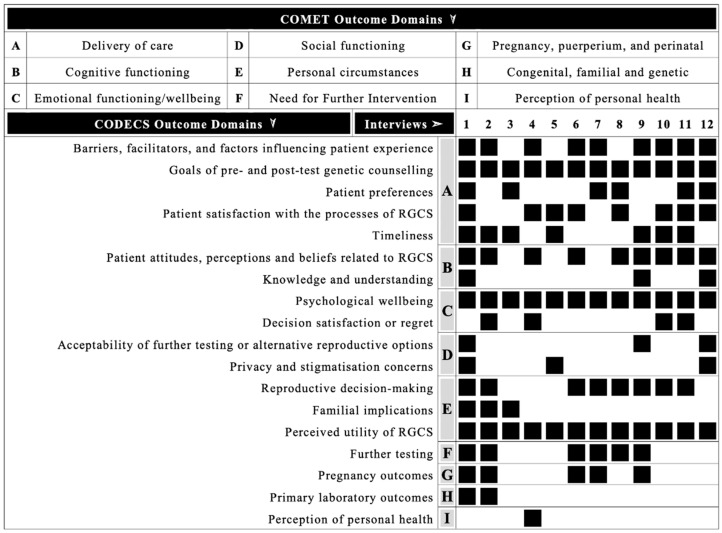
Block diagram illustrating the CODECS outcome domains represented across interviews, with higher level COMET taxonomy outcome domains indicated by A–I.

**Table 1 jpm-12-01310-t001:** Summary of participants.

Gender (*n* = 15)	Number of Participants
Female	12 (80%)
Male	3 (20%)
**Country (*n* = 12)**	
Australia	4 (33%)
Canada	1 (8%)
New Zealand (NZ)	2 (17%)
United Kingdom (UK)	1 (8%)
United States of America (USA)	4 (33%)
**Self-reported ethnicity (*n* = 15)**	
Ashkenazi Jewish	2 (13%)
Black African	1 (7%)
European New Zealand	2 (18%)
Multiracial (Hispanic, White and Native American)	1 (7%)
Multiracial (Aboriginal and Torres Strait Islander, White)	1 (7%)
White	8 (53%)
**Highest level of education (*n* = 15)**	
Vocational	2 (13%)
Tertiary—undergraduate	4 (27%)
Tertiary—postgraduate (Masters or PhD)	9 (60%)
**Timing of RGCS (*n* = 12)**	
Prenatal	2 (16%)
Preconception—proactive screening	5 (42%)
Preconception—following fetal loss	5 (42%)
**Type of RGCS (*n* = 12)**	
Expanded RGCS	11 (92%)
Midwife-led haemoglobinopathies screening	1 (8%)
**Risk Group (*n* = 12)**	
Low risk (no carrier findings)	1 (8%)
Low risk (one reproductive partner heterozygote for an autosomal recessive condition)	3 (25%)
Low risk (FXS premutation carrier with <1% risk of expansion)	1 (8%)
Increased risk couples identified through RGCS (female partner heterozygous for an X-linked condition, or both partners heterozygous for an autosomal recessive condition)	2 (16%)
Increased risk couples identified following fetal loss (no additional carrier findings on expanded RGCS)	5 (42%)

## Data Availability

The data presented in this study are available on request from the corresponding author. The data are not publicly available to maintain the privacy of participants.

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
