# Peer review of "Outcomes of Importance to Patients in Reproductive Genetic Carrier Screening: A Qualitative Study to Inform a Core Outcome Set"

_jpm, 2022, doi:10.3390/jpm12081310_

Round 1
Reviewer 1 Report
This manuscript describes one of the sub-studies of the CODECS study: an interview study to elicit outcomes of RGCS that are relevant to (prospective) parents.
The design of the study is robust and methods are well described, but I have questions about the added value/relevance of the outcomes of this study and feedback on the framing of the discussion. Therefore I recommend acceptation only after major revisions.
Grief and empowerment are stated to have not been described in other studies, but I would argue that this is 1) because of the characteristics of the study population, and 2) because informed decision making is central in most literature around RGCS, which is aimed at enabling empowerment. Furthermore they have presented to have found both the concepts of grief and empowerment already in their recently published literature review on the same topic (PMID: 35347269).
I hope to substantiate these concerns with my comments/questions below:
Introduction:
1) I would expect a reference to the published protocol for the CODECS study in the introduction (PMID: 34294124)
2) Throughout the manuscript (also in the title) participants are referred to as “patients”, but since also prospective parents with an average a priori risk were included it would perhaps be more appropriate to use other wording (e.g. “(prospective) parents”).
Although this might seem minor, I think it is important that the authors are clear that there is a distinction in expectations and experiences between those who are confronted with screening without (medical) indication in preconception setting and those who are undergoing carrier screening (e.g. to prevent another early pregnancy loss) in the IVF setting. This holds also true for presentation of the conclusions, in which no distinction is e.g. made between the outcomes relevant in different contexts/settings (preconception/proactive prenatal/IVF setting etc)
3) A nominal group technique is described to be adopted in 1-on-1 interviews (line 62, which seems appropriate and is well elaborated on in the method section), but it remains unclear why 1-on-1 interviews are executed and not e.g. focusgroups, to more directly aim towards consensus.
4) Aim 1 (line 71) is described as “ To explore the themes underlying participant interviews and how these inform our understanding of outcomes that are important to patients”. This to me seems like 2 aims: 1) To explore themes relevant to prospective parents undergoing RGCS, and 2) To understand why these themes are relevant to…..
Materials and methods:
1) In general I would like to express my compliments on the (description of the) design of the study
2) Please elaborate on why non-RGCS users were not eligible to participate in this study (line 88)? Could perhaps reasons not to choose RGCS also inform key outcomes?
3) What information did participants receive before the interview (e.g. in the social media posts referred to in line 105)? This could inform readers whether there could be any bias in the recruitment and/or responses.
Results:
1) Only 3 male participants are included (table 1): is it true that none of them were participating in a 1-on-1 interview (but all in couples)? What does that imply for the male role in reproductive decision making (and for further discussion: would you expect different outcomes if more males were interviewed)?
2) Table 1: please review the numbers in the table (e.g. the percentages don’t seem to add up to 100%)
3) Line 209: “All participants discussed the role of genetic counselling in promoting a sense of reproductive empowerment, by facilitating the provision of information regarding reproductive risk”: I would argue that genetic counseling is not merely provision of information, but also aids in comprehension of implications of (potential) reproductive risk and thereby promotes informed decision making, leading to (a sense of) reproductive empowerment.
Discussion:
1) Line 420-447: This paragraph frames the finding that Grief is an important potential (negative) outcome of RGCS. Although I agree that the concept of grief after RGCS could be further explored, I would argue that the loss of the sense of a “natural pregnancy” or medicalization are the main concepts that underlie the sense of grief in this context and these have been described elsewhere (e.g. PMID: 27388477 & PMID: 34112999), are also described by the authors in their systematic review (PMID: 35347269), and therefore are not new in this study.
Furthermore: is it possible (like described in the limitations, line 520) that overrepresentation of participants who had experienced fetal loss, may have made the outcome of grief (especially following the “tangible losses”, line 423) more in the forefront in these interviews/results and that this perhaps is not reflected in the more general population undergoing RGCS? I therefore question whether it should receive this much attention in the discussion and whether it should be framed as a new finding.
2) Line 469-495: This paragraph describes the importance of focus on (a feeling of) empowerment as outcome of RGCS. I would argue that, although participants might use different wording, the attention for e.g. enabling informed choice/decision making is underlying this concept/construct of empowerment and is studied as central outcome in different projects (as also described in PMID: 35347269). The authors could therefore perhaps elaborate on what aspect of other constructs they think would be included in what participants in this study considered empowerment and how this relates to the empowerment construct used for outcomes of genetic counseling by McAllistar et al. (PMID: 21255005).
The last sentence of this paragraph (line 492: “The clear conceptualization of utility encompassing both the clinical aspect of reproductive decisions and the personal aspect of empowerment will ensure a truly patient-informed and ethically –oriented definition of utility for the purpose of a core outcome set.”) is furthermore up to debate: many would argue that clinical aspects should not be the primary aim, especially in the context of screening the “general population”. This might however also be very culturally dependent, as e.g. described in PMID: 31833001.
Reviewer 2 Report
The article is dedicated to the actual problem of modern medical-genetic counseling, namely, people's perception of the results of carrier tests. The authors propose an original method of interviewing and analyzing the results. Thanks to the proposed methodology, it was possible to identify previously hidden points of interest from the attention of other researchers. Along with this, it is necessary to note the critically small size and bias of the sample (the bias is obviously caused by the selection of respondents, including through groups of parents of patients). The study compares two groups, which makes their size quite insignificant for evaluating such a large number of parameters. And I recommend that authors with such a sample size use absolute numbers and not percentages and fractions.
I think the work is certainly interesting, but I recommend reducing it to a brive report, describing the methodology and presenting the results as a test to test the methodology, which showed its success and the possibility of further use.
